# Association between Migraine and the Risk of Stroke: A Bayesian Meta-Analysis

**Kim-Ngan Ta-Thi** [1,2]**, Kai-Jen Chuang** [1,3,*] **and Chyi-Huey Bai** [1,3]

1   School of Public Health, College of Public Health, Taipei Medical University, Taipei 110, Taiwan; d508108005@tmu.edu.tw (K.-N.T.-T.); baich@tmu.edu.tw (C.-H.B.)
2   Faculty of Public Health, University of Medicine and Pharmacy at Ho Chi Minh City, Ho Chi Minh City 700000, Vietnam
3   Department of Public Health, School of Medicine, College of Medicine, Taipei Medical University, Taipei 110, Taiwan
*   Correspondence: kjc@tmu.edu.tw; Tel.: +886-2-2736-1661

**Abstract:** There are still inconsistent results about association between migraine and stroke risk in studies. This paper was to review findings on the association between migraine (with or without aura) and stroke risk. We searched articles in the Embase and PubMed up to January 2021. Two independent reviewers extracted basic data from individual studies using a standardized form. Quality of studies was also assessed using the Newcastle–Ottawa Scale. We conducted a meta-analysis, both classical and Bayesian approaches. We identified 17 eligible studies with a sample size more than 2,788,000 participants. In the fixed effect model, the results demonstrated that migraine was positively associated with the risk of total stroke, hemorrhagic stroke, and ischemic stroke. Nevertheless, migraine was associated with only total stroke in the random effects model (risk ratio (RR) 1.31, 95%CI: 1.06–1.62). The probability that migraine increased total stroke risk was 0.978 (RR 1.31; 95% credible interval (CrI): 1.01–1.72). All types of migraine were not associated with ischemic stroke and hemorrhagic stroke. Under three prior distributions, there was no association between migraine and the risk of ischemic stroke or hemorrhagic stroke. Under the non-informative prior and enthusiastic prior, there was a high probability that migraine was associated with total stroke risk.

**Keywords:** migraine; ischemic stroke; hemorrhagic stroke; meta-analysis; Bayesian; headache

## 1. Introduction

Migraine and other headache disorders have become one of the leading causes of disability since the Global Burden of Diseases, Injuries, and Risk Factors studies in 2000 produced burden estimation of migraine worldwide. The results of Global Burden of Diseases, Injuries, and Risk Factors studies in 2016 also showed that headache, especially migraine, is a major global health issue in both male and female among all age groups worldwide [1]. This affects not only employees' health but also their productivity in the labor sector, which acts as a deterrent for sustainable development of organizations and countries.

According to Global and regional burden of disease and risk factors in 2001, cerebrovascular accidents (stroke) is the second leading cause of death and one of the ten leading causes of disease burden (measured in disability-adjusted life years—DALYs) in both high-income countries and low-and-middle-income countries [2]. There are a lot of risk factors related to stroke such as high blood pressure, older age, heart disease, high cholesterol, smoking, atherosclerosis, diabetes, and a family history of stroke [3]. In addition, the association between migraine and stroke has also received attention from many researchers because preventive strategies to reduce the number of stroke patients based on migraine status contribute to not only economic burden reduction of stroke for

public health systems, but also sustainable development at the individual, community, and national level.

From 2004 to present, four previous meta-analyses on the association between migraine and stroke and one previous meta-analysis on the association between migraine and cardiovascular diseases drew inconsistent conclusions [4–8]. The latest meta-analysis of 18 cohort studies in 2017 found that migraine was associated with long-term risk of cerebrovascular events (both ischemic and hemorrhagic stroke) and cardiovascular events [5]. Since the publication date of that meta-analysis, several additional studies have examined the association between migraine and stroke and yielded contradictory findings with it [9–11]. Hence, a better understanding of association between stroke and migraine in general population is needed to take steps to reduce brain health problems in the Sustainable Development Goals.

In this paper, we used both classical or frequentist (fixed effects and random effects models) meta-analysis and Bayesian meta-analysis. The classical approach, also frequentist statistics (proof of contradiction), has its own limitation in that the *p*-value is sensitive to sample size or sample size has an effect on the credibility of study results [12]. In addition, approximate 25% of the findings with a *p*-value less than 0.05 yielded false positive results [13,14]. Contrasting with the classical approach, the Bayesian approach applied the well-known Bayes theorem [15,16]. This approach does not depend on either sample size or *p*-value [17]. This inference method is an updated process by forming the "posterior information" based on "prior information" and the existing data from the study. In the frequentist approach, the *p*-value is the probability of the observed outcome given that the null hypothesis is true (Ho: there is no relationship between migraine and stroke), denoted by P (data observed | alternative hypothesis), or confidence interval is interpreted as 95% of random samples of study subjects will produce confidence intervals that contain the true proportion of migraine is associated with the risk of stroke. In contrast, Bayesian statistics aim to answer the question: Given the data we observed, what is the probability that some event of interest happens (i.e., the probability that migraine increases stroke risk at least 10%), which is denoted by P (event of interest | data observed). Therefore, the purpose of this study was to conduct a meta-analysis using the classical and Bayesian approaches to update the association between migraine and the risk of stroke.

## 2. Materials and Methods

### 2.1. Search Strategy and Selection Criteria

The studies included in this analysis were from two sources: five previous meta-analyses and newly identified studies. The five previously published meta-analyses identified 72 studies. We then conducted further searches for new studies that had been published since the publication of the latest meta-analysis in 2017. This search was conducted until January 2021 using the Embase and PubMed (during the last four years) with the following keywords: "stroke", "migraine", "cardiovascular diseases", "headache", and "Cerebrovascular accident".

The papers were included in the analysis based on the following criteria: (1) written in the English language; (2) studies on the human; (3) studies with clearly defined stroke as an outcome; (4) original papers; (5) studies reported risk ratio (RR) and its 95% confidence interval (CI); (6) cohort study design; and (7) migraine as the exposure of interest. In case of duplicate publications, the first published paper with data on the number of migraine and stroke cases were included. Studies would be excluded if data on migraine and stroke were not available and corresponding authors could not be contacted. In this paper, we included only cohort studies to evaluate long-term effects of migraine on stroke of large groups of individuals.

### 2.2. Data Analysis

Data Extraction and Quality Assessment

Two reviewers (K.-N.T.-T. and K.-J.C.) independently examined papers or abstracts to extract basic data onto a standardized form. Any discrepancy between the two reviewers was resolved by verification of the third reviewer (C.-H.B).

The Newcastle–Ottawa Scale, a validated scale, was used to assess the quality of nonrandomized studies (case-control and cohort studies) for each study [18]. In this scale, the cohort studies were assessed by three subscales: (a) selection, (b) comparability, and (c) outcome. Each subscale was rated maximum four, two, and three "stars" as scores respectively based on defined criteria [19]. A study with a total score of at least 7 was considered a high quality study [5].

### 2.3. Meta-Analysis

Classical meta-analysis was performed in both fixed-effect and random-effects models. In the Bayesian random-effects meta-analysis, we estimated the logarithm of relative risk-RR (denoted by $\theta$) of each study and its variance (denoted by $\sigma^2$). The result of this estimation was assumed to be normally distributed with overall effect of log RR for these studies (denoted by $\mu$) and between-study variability $\tau^2$. Here, prior distribution for $\mu$, $\sigma^2$ and $\tau^2$ must be specific. The prior distribution for $\tau^2$ was assumed to be uniformly distributed with parameters (0, 10). Based on the Bayes theorem, we used three prior distributions below for $\mu$ and $\sigma^2$ as described elsewhere [15,20,21]:

Non-informative prior: It is hypothesized that the probabilities of migraine have a positive affect or a negative effect on stroke are equally likely. Therefore, overall RR was set with average 1, which means no effect of migraine on stroke (or average $\mu = 0$) and variance as large as 10,000.

Sceptical prior (centered on a "null" value for risk ratio with a reasonable expression of doubt about the overall effect): It is thought that there is a small probability (i.e., 5%) that migraine might reduce by more than 50% stroke risk (RR $\leq 0.5$) or increase by more than 50% stroke risk (RR $\geq 1.5$). In logarithmic scale, it is equivalent to P($\mu \leq -0.693$) = 0.05 and P($\mu \geq 0.693$) = 0.05, respectively. Therefore, overall RR was set with average 1 ($\mu = 0$) and the prior variance was calculated as $(0.693/1.645)^2 = 0.177$.

Enthusiastic prior (opposite of the sceptical prior): It is assumed that migraine might increase by 50% risk of stroke (RR $\geq 1.5$) with the same precision as the sceptical prior. Hence, $\mu$ was set at a mean of 0.693 and variance of 0.177.

Bias might occur in observational studies. It is assumed that potential bias causes observed RR in each study from 50% lower to 50% higher than the true RR. Therefore, the average difference between the observed RR and the true RR is 0 and variance as $(\log(1.5)/1.96)^2 = 0.0427$. Specifically, in this sensitivity analysis, we assumed overestimation of the true RR from 10% to 30% and kept the variance in a constant value 0.0427. Bayesian sensitivity analysis was performed using the Markov Chain Monte Carlo simulations as described elsewhere [21].

Based on three prior distributions, we estimated the probability that RR is more than 1, 1.1, and 1.2. The posterior distributions and 95% credible intervals (CrI) were reported. Inferences were run 10,000 iterations after discarding the first 1000 (burn-in) on one chain to reach the convergence. $I^2$ (index of heterogeneity across studies or coefficient of inconsistency) is considered as low, moderate, and high degrees of heterogeneity if the $I^2$ value is over 25%, 50%, and 75%, respectively [22]. The funnel plots and the Egger tests were used to identify publication bias in the included studies. All statistical analyses were performed in OpenBUGS version 3.2.3 (Cambridge, UK) and R version 4.0.2 (R Foundation, Australia).

### 3. Results

*3.1. Characteristics of Included Studies*

Seventeen studies were included in this Bayesian meta-analysis (Figure 1). Most of the studies were based on large populations, with eight studies including men and women. Only five studies reported both hemorrhagic and ischemic stroke and eight studies reported types of migraine in detail (with or without aura). A total of more than 2,788,000 participants were included in this analysis. Most of the studies have a sample size larger than 10,000 participants. Fourteen studies were evaluated "high quality" with quality scores at least 7 (Table 1).

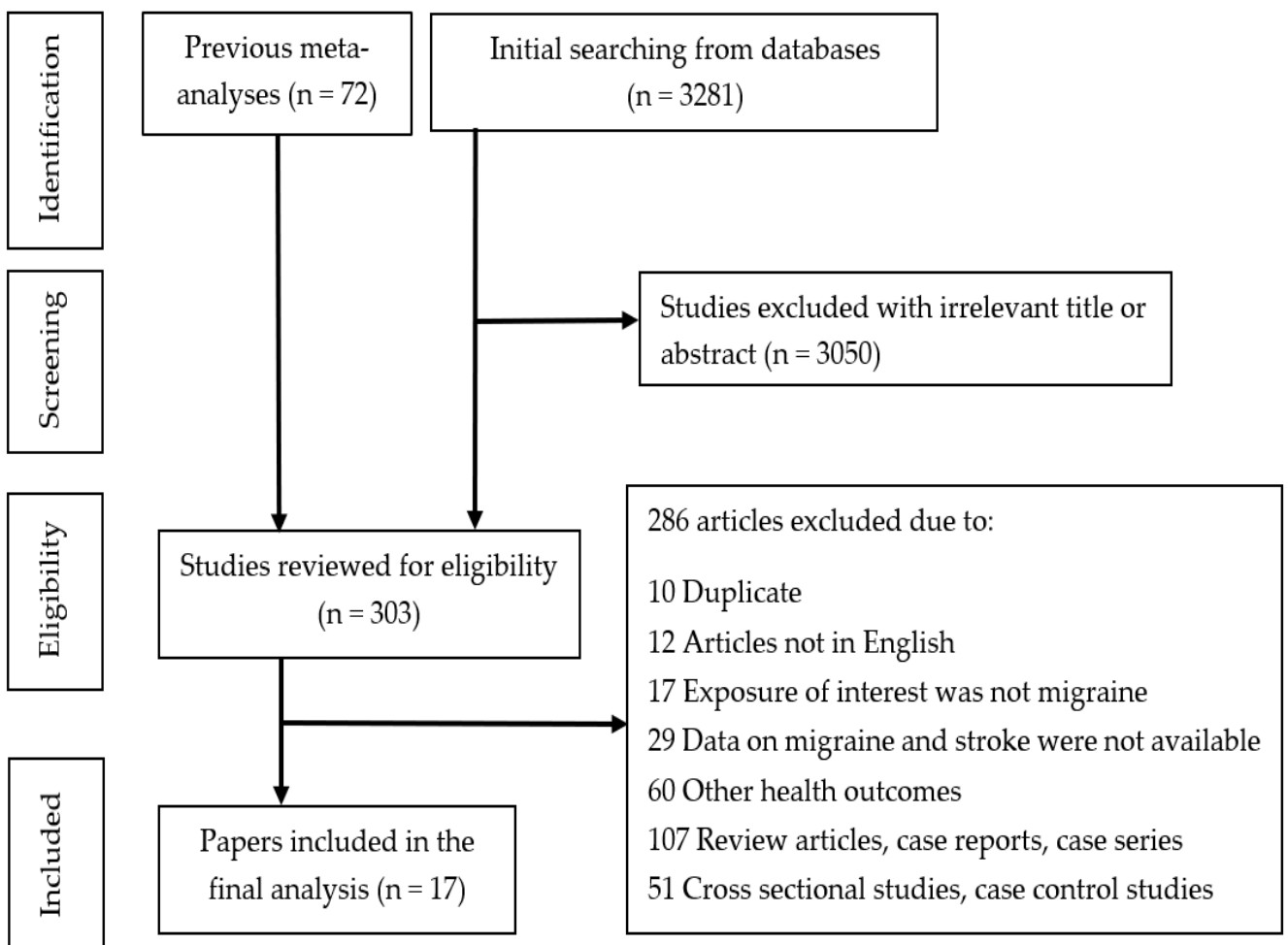

**Figure 1.** Number of studies identified, included, and excluded from analysis.

**Table 1.** Characteristics of included studies.

| No | First Author | Year | Follow-Up Years | Gender | Age (Mean or Range) | Territory | Sample Size | Quality Score | Type of Migraine | Type of Stroke |
|----|------|------|------|------|------|------|------|------|------|------|
| 1 | Buring [23] | 1995 | 5 years | Men | 40–84 | USA | 21,960 | 7 | Any | Total, hemorrhagic, ischemic |
| 2 | Merikangas [24] | 1997 | Not Applicable | Both | 25–74 | USA | 12,090 | 7 | Any | Total |
| 3 | Hall [25] | 2004 | 2.9 years | Both | ≥15 | UK | 140,814 | 8 | Any | Total, hemorrhagic, ischemic |
| 4 | Velentgas [26] | 2004 | 1.4 years | Both | 38 | USA | 260,822 | 8 | Any | Total |
| 5 | Kurth [27] | 2005 | 9 years | Women | ≥45 | USA | 39,754 | 7 | With/without aura | Total, hemorrhagic, ischemic |
| 6 | Kurth [28] | 2006 | 10 years | Women | ≥45 | USA | 27,840 | 7 | With/without aura | Ischemic |
| 7 | Kurth [29] | 2007 | 15.7 years | Men | 40–84 | USA | 20,084 | 7 | Any | Ischemic |
| 8 | Kurth [30] | 2010 | 13.6 years | Women | ≥45 | USA | 27,860 | 6 | With/without aura | Hemorrhagic |
| 9 | Kuo [31] | 2013 | 2 years | Both | 43 | Taiwan | 125,550 | 8 | With/without aura | Hemorrhagic |
| 10 | Gelfand [32] | 2015 | 10 years | Children | 2–17 | USA | 1,411,306 | 6 | Any | Total, hemorrhagic, ischemic |
| 11 | Kurth [33] | 2016 | 20 years | Women | 25–42 | USA | 115,541 | 8 | Any | Total |
| 12 | Peng [34] | 2016 | 3.6 years | Both | 41 | Taiwan | 238,124 | 7 | With/without aura | Ischemic |
| 13 | Androulakis [35] | 2016 | 20 years | Both | 59 | USA | 12,758 | 8 | With/without aura | Ischemic |
| 14 | Rambarat [36] | 2017 | 4 years | Women | 58 | USA | 917 | 6 | Any | Total |
| 15 | Lantz [37] | 2017 | 11.9 years | Both | 45.3 | Sweden | 53,404 | 7 | With/without aura | Total, hemorrhagic, ischemic |
| 16 | Lee [9] | 2019 | 6.7 years | Both | ≥20 | South Korea | 207,925 | 9 | With/without aura | Total, hemorrhagic, ischemic |
| 17 | Pavlovic [11] | 2019 | 22 years | Women | 50–79 | USA | 71,441 | 7 | Any | Total |

### 3.2. Association between Migraine and Risk of Total Stroke, Hemorrhagic Stroke, and Ischemic Stroke (Classical Meta-Analysis)

The observed RR and 95% CI for total stroke, hemorrhagic stroke, and ischemic stroke in included studies and overall RR are shown in Figures 2–4. In fixed effect model, three overall RR consistently showed that migraine was positively associated with the risk of total stroke, hemorrhagic stroke, and ischemic stroke. However, migraine was associated with only total stroke in the random effects model. Significant heterogeneity was observed for total stroke, hemorrhagic stroke, and ischemic stroke with $I^2 > 80\%$ ($p < 0.01$).

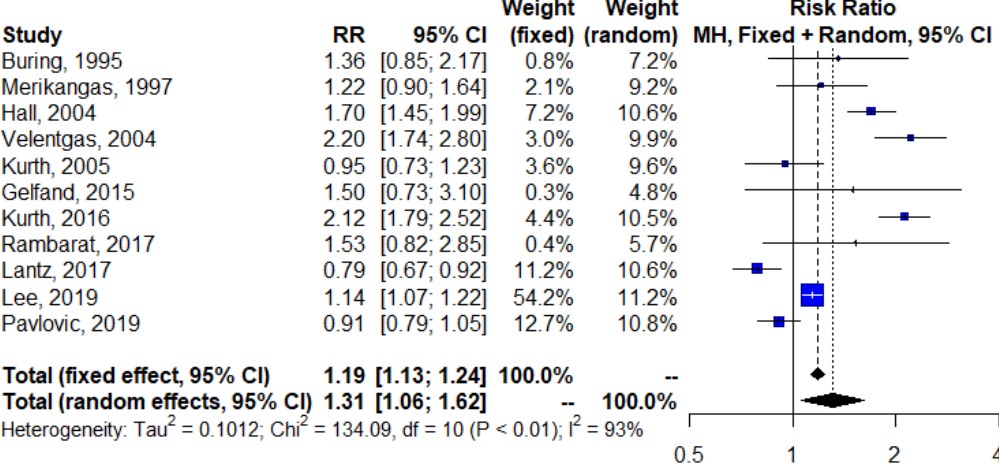

**Figure 2.** Forest plot of risk ratio for migraine and total stroke.

**Figure 3.** Forest plot of risk ratio for migraine and ischemic stroke.

**Figure 4.** Forest plot of risk ratio for migraine and hemorrhagic stroke.

### 3.3. Migraine with Aura or without Aura and Stroke (Or Subgroup Analysis)

Table 2 presents results of subgroup analysis by types of migraine. All types of migraine were not associated with ischemic stroke and hemorrhagic stroke compared to non-migraine. There was evidence from moderate heterogeneity to high heterogeneity among studies in with $I^2 > 60$ ($p < 0.05$).

**Table 2.** Subgroup analysis by types of migraine.

| Subgroup | Ischemic Stroke | | | Hemorrhagic Stroke | | |
|---|---|---|---|---|---|---|
| | RR (95% CI) | *p* Value | $I^2$ (%) (*p*) | RR (95% CI) | *p* Value | $I^2$ (%)(*p*) |
| Overall (any migraine) | 1.10 (0.93–1.30) | 0.25 | 84.2 ($p < 0.0001$) | 1.11 (0.79–1.55) | 0.55 | 83.6 ($p < 0.0001$) |
| Migraine with aura | 2.03 (0.84–4.88) | 0.11 | 99.1 ($p < 0.0001$) | 1.23 (0.78–1.93) | 0.38 | 61.2 ($p = 0.0355$) |
| Migraine without aura | 1.32 (0.76–2.30) | 0.32 | 98.6 ($p < 0.0001$) | 1.11 (0.79–1.56) | 0.56 | 61.2 ($p = 0.0356$) |

### 3.4. Publication Bias

The funnel plots (Figures 5–7) show a symmetry for total stroke ($p = 0.46$), ischemic stroke ($p = 0.93$), and hemorrhagic stroke ($p = 0.69$), indicating no significant publication bias.

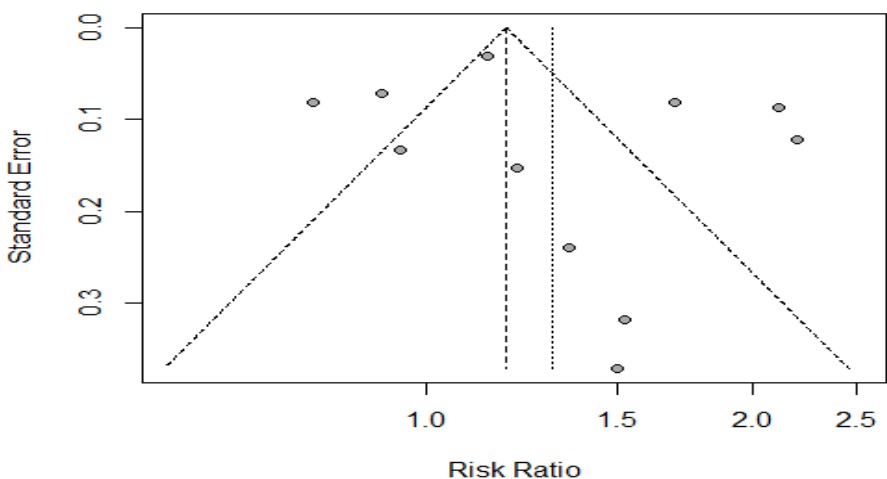

**Figure 5.** Funnel plot for total stroke.

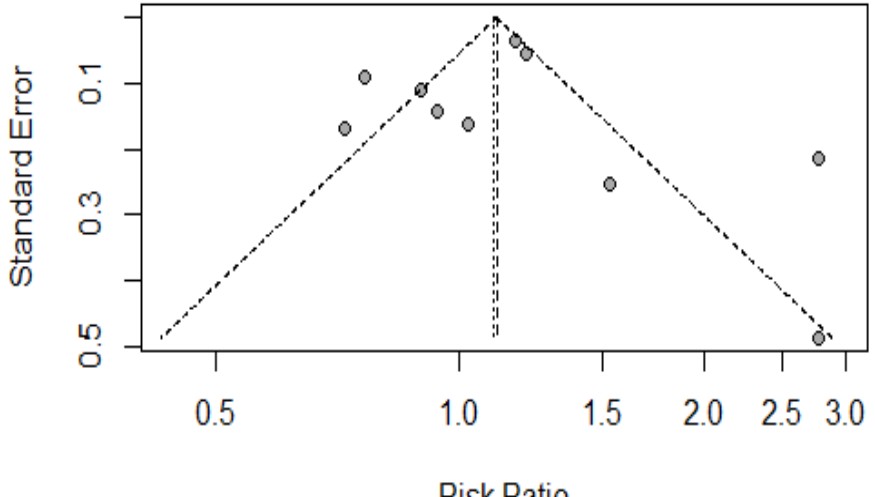

**Figure 6.** Funnel plot for ischemic stroke.

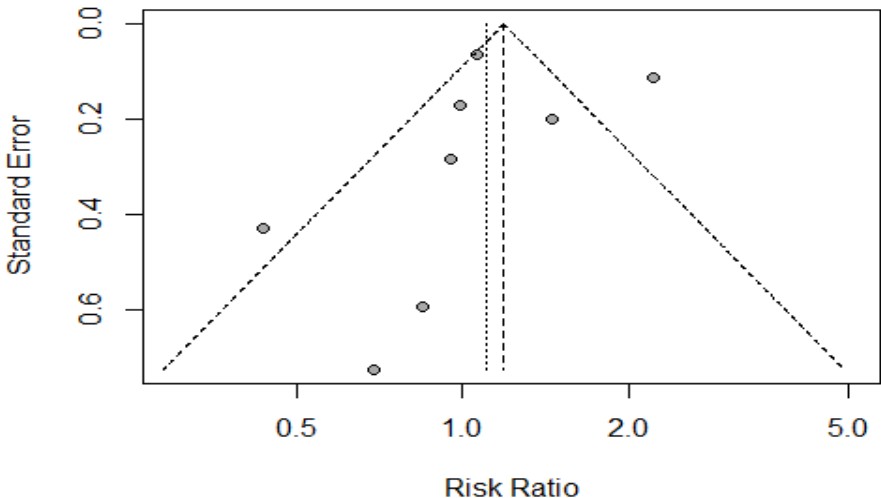

**Figure 7.** Funnel plot for hemorrhagic stroke.

*3.5. Overall Risk Ratio and 95% Credible Interval under 3 Priors (Bayesian Meta-Analysis)*

Under the three prior distributions, the results of Table 3 showed risk ratio, 95% CrI, and probability of risk ratio (the probability that migraine increases stroke risk by at least 10%, 20%, and 30%, respectively). Under the non-informative prior, the overall RR for total stroke was 1.31 (95% CrI: 1.01–1.72). The probability that migraine increases stroke risk was 97.8%. The probability that migraine increases stroke risk by at least 10% and 20% was 91.8% and 76.8%, respectively.

**Table 3.** Overall risk ratio and 95% credible interval for stroke under three priors.

| Stroke Type | RR (95% CrI) | Probability (%) That Risk Ratio | | |
|---|---|---|---|---|
| | | >1.0 | >1.1 | >1.2 |
| **Non-informative prior** | | | | |
| Total stroke | 1.31 (1.01–1.72) | 97.8 | 91.8 | 76.8 |
| Ischemic stroke | 1.15 (0.84–1.60) | 83.3 | 61.0 | 36.7 |
| Hemorrhagic stroke | 1.09 (0.67–1.70) | 67.8 | 50.3 | 33.1 |
| **Sceptical prior** | | | | |
| Total stroke | 1.28 (1.00–1.64) | 97.3 | 89.7 | 71.8 |
| Ischemic stroke | 1.13 (0.85–1.51) | 82.1 | 57.3 | 31.6 |
| Hemorrhagic stroke | 1.08 (0.72–1.56) | 67.2 | 47.2 | 28.4 |
| **Enthusiastic prior** | | | | |
| Total stroke | 1.36 (1.06–1.76) | 99.3 | 96.1 | 85.5 |
| Ischemic stroke | 1.23 (0.92–1.71) | 92.8 | 76.8 | 53.5 |
| Hemorrhagic stroke | 1.24 (0.84–1.83) | 88.1 | 74.1 | 55.6 |

RR: risk ratio; 95% CrI: 95% credible interval.

Compared to the non-informative prior, these results were indifferent significantly when the prior distribution was enthusiastic prior. Under the sceptical prior, migraine was not associated with total stroke, ischemic stroke, and hemorrhagic stroke.

*3.6. Sensitivity Analysis of Bias*

Table 4 demonstrates results of sensitivity analysis on the bias of the relationship between migraine and stroke risk under non-informative prior. If 10% bias occurred in each study, the probability of a positive effect (RR > 1) of migraine decreased to 92.3%, 76.1%, 20.7% for total stroke, ischemic stroke, and hemorrhagic stroke, respectively. The probability of an effect would decrease further when bias increases from 10% to 30% in any type of stroke.

**Table 4.** A Bayesian sensitivity analysis on the bias of association between migraine and stroke risk under non-informative prior.

| Stroke Type | Bias | RR (95% CrI) | Probability (%) That Risk Ratio | | |
| --- | --- | --- | --- | --- | --- |
| | | | >1.0 | >1.1 | >1.2 |
| Total stroke | 10% | 1.13 (0.95–1.35) | 92.3 | 63.5 | 25.7 |
| | 20% | 1.01 (0.85–1.20) | 53.4 | 16.0 | 2.5 |
| | 30% | 0.88 (0.74–1.05) | 7.7 | 0.8 | 0.1 |
| Ischemic stroke | 10% | 1.11 (0.82–1.49) | 76.1 | 51.9 | 29.0 |
| | 20% | 0.98 (0.73–1.33) | 45.4 | 22.4 | 8.9 |
| | 30% | 0.86 (0.64–1.16) | 15.4 | 5.0 | 1.7 |
| Hemorrhagic stroke | 10% | 0.88 (0.63–1.22) | 20.7 | 8.4 | 3.1 |
| | 20% | 0.78 (0.56–1.09) | 6.7 | 2.3 | 0.9 |
| | 30% | 0.68 (0.49–0.95) | 1.5 | 0.5 | 0.2 |

## 4. Discussion

The association between migraine and stroke remains controversial. In this study, overall, the risk of total stroke increased 31% among migraineurs. The observed effect size of migraine in this analysis in any prior distribution is lower than those observed in the previous meta-analyses [5,7], which reported a relative risk of 53% and 55% increase in total stroke, respectively. In contrast to previous meta-analyses [4–7], with accumulative data, the effect size for ischemic stroke in this study shows there was no effect of migraine in any prior distributions. The current results in any prior information that migraine was not associated with hemorrhagic stroke are consistent with Hu's study [7] and not consistent with two other previous findings [5,8]. This present study excluded case-control studies, which might be influenced by recall bias and more cohort studies with large sample size and long-term follow-up duration increase higher evidence, suggesting a true effect of migraine on stroke for both sexes and all ages. The number of original included studies and inclusion criteria might also cause different findings between this study and previous meta-analyses.

The underlying mechanism for how migraine affects stroke is still not clear. There are some mechanisms that might explain this association. "Cortical spreading depression", a pathophysiologic mechanism of migraine with aura, may lead to an ischemic stroke [38]. Increased platelet aggregation, alterations in endothelial function, platelet activation, and von Willebrand factor might relate to the link between migraine and stroke [39–41]. In addition, young stroke patients with migraine had higher frequency of hypercoagulable states [42].

Bayesian meta-analysis results under prior distributions except sceptical prior were consistent with those of classical meta-analysic. In addition, there is a high chance of 76.8% that migraine is associated with more than 20% increase in total stroke risk under non-informative prior. Results in Table 4 also showed that the link between migraine and any types of stroke no longer exists if 10% bias occurred in each study in the sensitivity analysis. Furthermore, the probability of migraine associated to stroke decreases significantly when bias increases from 10% to 30% in any type of stroke. This bias might be caused by inaccurate methods to ascertain migraine patients and stroke cases such as interviews or one self-reported question in included studies.

Although the current analysis supports the hypothesis that migraine increases risk of total stroke in the non-informative prior and enthusiastic prior, these findings must be considered in terms of strengths and weaknesses. The current analysis with more than 2,788,000 subjects can estimate effect size more precisely than results from individual studies. In addition, the Bayesian approach in this study has more advantages than the classical meta-analysis. Firstly, it is possible to define directly the probability that the hypothesis is true, which is impossible in frequentist statistics [43]. Secondly, effects of bias can be considered in the Bayesian approach [44].

However, the quality of meta-analysis depends on the quality of included studies [45]. This limitation might not enable our study to control confounders because it was based on observational studies. Stroke could relate to many risk factors that can be controlled

better in randomized controlled trials such as high blood pressure, older age, heart disease, high cholesterol, smoking, atherosclerosis, diabetes, and a family history of stroke [3]. In addition, some included cohort studies were not adequate in terms of loss of follow up, or no description of those lost, which might affect the outcome results.

Moreover, because types of stroke in some included studies were not specified, the results of the subgroup analysis and publication bias for these studies might not be accurate. In addition, effects of types of migraine (with or without aura), frequency of migraine, and migraine age of onset on stroke were not defined in most included papers. Therefore, the association between these factors and stroke should be addressed in future studies.

## 5. Conclusions

In conclusion, this current analysis shows that the risk of total stroke increased 31% among migraineurs. The probability that migraine increases stroke risk was 97.8%. The regular health examinations with focus on cerebrovascular symptoms should be a part of personal preventive health care strategy and/or a part of social welfare system. The risk of ischemic stroke or hemorrhagic stroke in migraineurs and association between stroke and types of migraine (with or without aura), frequency of migraine, and migraine age of onset should be further investigated in the future studies.

**Author Contributions:** Conceptualization, methodology, literature search, K.-N.T.-T., K.-J.C.; data analysis, K.-N.T.-T., C.-H.B.; writing—original draft preparation, K.-N.T.-T.; writing—review and editing, K.-J.C. All authors have read and agreed to the published version of the manuscript.

**Funding:** This research received no external funding.

**Institutional Review Board Statement:** Not applicable.

**Informed Consent Statement:** Not applicable.

**Data Availability Statement:** Not applicable.

**Conflicts of Interest:** The authors declare no conflict of interest.

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
