# Peer review of "Association between Migraine and the Risk of Stroke: A Bayesian Meta-Analysis"

_sustainability, doi:10.3390/su13073759_

Round 1

Reviewer 1 Report

Association between migraine and the risk of stroke: A Bayesian meta-analysis

Peer Review

Overall, a good paper with academic and applied/practical contributions, especially to data selection, statistical analysis, and overall methodology. However, the paper could use a little more “meat” (plenty of room to expand on some issues) and brief definitions and explanations to expand audience impact and include a readership less familiar with various aspects of the study, especially the stats. Also, the paper will need some basic English editing for word/phrase choice, grammar, some sentence structuring, etc. The overall organization is adequate.

Below are some editorial suggestions. I only provide a basic quick list of suggestions for the first half of the paper. The paper will need more careful attention, but most of the editorial needs are relatively minor. The English is clear and fairly well written for presumably non-native English speakers. Also, I remind the authors that these are merely suggestions. The authors are free to use or ignore suggestions at their discretion. I may have misinterpreted some things as well. Apologies if this is the case.

Line 29: Not sure if I would say “important causes”. “Primary causes” or “concerning causes”… followed by it’s an “important concern” or “critical concern”.

Line 33: “… among all age groups…”

Line 34: “… For example, in the labor sector, this affects not only…”

Line 42: “…has also received attention…”

Line 43 “… health systems…” Overall, needs some English editing; also, some grammatical, word/phrase choice and other editing for better readability.

Line 53: “… these studies have…”

Line 57: “… sample size which means…”

Line 58-65: “…approximately 25% of the findings… yielded false positive results… Contrasting with… does not depend… is an updated process…”

This section is OK. The purpose is clearly stated in lines 63-64. The study is also relevant and important – for both academic and applied/practical sectors. The statistical approaches, data sets, and methodology are good contributions to methodological concerns in research (i.e., important contribution).

Needs some basic English editing by native English speaker and professional semi-familiar with the field of study. However, the English is fairly good and editing needs are minor.

Line 70: “… conducted further searches…”

Line 72: “… until January…”

I am going to cease with minor editorial suggestions here. However, the rest of the document needs similar editorial attention.

Section 2: The data/database and study selection – part of the methodology and analytical/statistical methods are clearly stated. Some further explanatory textual information on the pros/cons and why these particular statistical methods were used may help readers less familiar with the statistical approaches.

Line 107: “skeptical prior” ? Confusing. Reword and explain. Same with “optimistic prior”.

Table 1: Definitely a US bias. What are the implications? For example, there are cultural factors such as food and eating cultures/habits that may differ from one culture to the next; also genetic factors that may relate to ethnicity; and socio-economic and other demographic factors such as income (e.g., low income Americans cannot afford to eat healthy; work stresses and sleeping habits may differ dramatically; work environments (healthy, toxic) may differ dramatically; time/access to proper exercise and rest habits; access to medical attention… and so forth… to also include ‘genetic’ predispositions and existing medical conditions of individuals which may be related to stroke, stroke risk either directly and/or a cumulative risk increase… somewhat discussed in lines 214-220 and 229-234). Do any studies consider these factors. Petting all this “big data” demographic and cultural information into at least correlation analyses might be useful, but not entirely necessary for the purposes here. However, the authors may wish to bring up some of these concerns for future studies.

Lines 177-185: Good information here.

Table 4: “Probably (%) that risk ratio…???” [risk ratio what? Need to explain more clearly what this column(s) means]. Also, forgot to mention above that terms such as “equivocal prior, skeptical prior and optimistic prior” may need rewording and brief simple definitions should be provided for each to increase readership and understanding among those less familiar with terms and expressions.

Lines 221-228: Good points.

Lines 229-242: Good points on some of the limitations, gaps, biases, and other considerations. This is useful to know for future research design in order to reduce some of these concerns.

Section 5 (Conclusion) is a bit brief. More could be added, including “way forward” or recommendations for future research.

Author Response

Q1. Overall, a good paper with academic and applied/practical contributions, especially to data selection, statistical analysis, and overall methodology. However, the paper could use a little more “meat” (plenty of room to expand on some issues) and brief definitions and explanations to expand audience impact and include a readership less familiar with various aspects of the study, especially the stats. Also, the paper will need some basic English editing for word/phrase choice, grammar, some sentence structuring, etc. The overall organization is adequate.Below are some editorial suggestions. I only provide a basic quick list of suggestions for the first half of the paper. The paper will need more careful attention, but most of the editorial needs are relatively minor. The English is clear and fairly well written for presumably non-native English speakers. Also, I remind the authors that these are merely suggestions. The authors are free to use or ignore suggestions at their discretion. I may have misinterpreted some things as well. Apologies if this is the case.

R1. We truly appreciate your comments. We definitely revise our manuscript according to reviewers’ comments.

Q2. Line 29: Not sure if I would say “important causes”. “Primary causes” or “concerning causes”… followed by it’s an “important concern” or “critical concern”.

R2. We reworded “important causes” into “leading causes”

Q3. Line 33: “… among all age groups…”

R3. We reworded “at all age” into “among all age groups”

Q4. Line 34: “… For example, in the labor sector, this affects not only…”

R4. We revise that question: “This affects not only on employees’ health but also their productivity in the labor sector which acts as a deterrent for sustainable development of organizations and countries.

Q5. Line 42: “…has also received attention…”

We wrote this sentence in passive voice “…has also been received attention by many researchers …”

Q6. Line 43 “… health systems…” Overall, needs some English editing; also, some grammatical, word/phrase choice and other editing for better readability.

R6. We reworded “health system” into “health systems”

Q7. Line 53: “… these studies have…”

R7. We combine 2 sentences into “Since the publication date of that meta-analysis, several additional studies have examined the association between migraine and stroke and yielded contradictory findings with it”

Q8. Line 57: “… sample size which means…”

R8. We deleted “which means”

Q9. Line 58-65: “…approximately 25% of the findings… yielded false positive results… Contrasting with… does not depend… is an updated process…”

R9. We revised as all above words and deleted “a chance of”

Q10. This section is OK. The purpose is clearly stated in lines 63-64. The study is also  relevant and important – for both academic and applied/practical sectors. The statistical approaches, data sets, and methodology are good contributions to methodological concerns in research (i.e., important contribution). Needs some basic English editing by native English speaker and professional semi- familiar with the field of study. However, the English is fairly good and editing needs are minor.

R10. We appreciate your valuable comments.

Q11. Line 70: “… conducted further searches…”; Line 72: “… until January…”

R11. We reworded “search” into “searches” and “up to” into “until”

Q12. I am going to cease with minor editorial suggestions here. However, the rest of the document needs similar editorial attention. Section 2: The data/database and study selection – part of the methodology and analytical/statistical methods are clearly stated. Some further explanatory textual  information on the pros/cons and why these particular statistical methods were used may help readers less familiar with the statistical approaches.

R12. To clarify 2 statistical approaches, we added this in Introduction “In frequentist approach, p-value is the probability of the observed outcome given that the null hypothesis is true (Ho: there is no relationship between migraine and stroke), denoted by P(data observed | alternative hypothesis), or confidence interval is interpreted as 95% of random samples of study subjects will produce confidence intervals that contain the true proportion of migraine is associated with the risk of stroke. In contrast, Bayesian statistics aims to answer the question: Given the data we observed, what is the probability for some event of interest happens (i.e. The probability that migraine increases stroke risk at least 10%)?, which is denoted by P(event of interest | data observed).

Q13. Line 107: “skeptical prior” ? Confusing. Reword and explain. Same with “optimistic   prior”.

R13. I added 3 reference to clarify how to build prior distribution and rewrote 3 priors as follows:

“Non-informative prior: It is hypothesized that the probabilities of migraine have a positive affect or a negative effect on stroke are equally likely. Therefore, overall RR was set with average 1 which means no effect of migraine on stroke (or average μ = 0) and variance as large as 10,000.

Sceptical prior (centred on a “null” value for risk ratio with a reasonable expression of doubt about the overall effect): It is thought that there is a small probability (i.e. 5%) that migraine might reduce by more than 50% stroke risk (RR ≤ 0.5) or increase by more than 50% stroke risk (RR ≥ 1.5). In logarithmic scale, it is equivalent as P(μ ≤ − 0.693) = 0.05 and P(μ ≥ 0.693) = 0.05, respectively. Therefore, overall RR was set with average 1 (μ = 0) and the prior variance was calculated as (0.693/1.645)2 = 0.177.

Enthusiastic prior (the opposite of the sceptical prior): It is assumed that migraine might increase by 50% risk of stroke (RR ≥ 1.5) with the same precision as the sceptical prior. Hence, μ was set at a mean of 0.693 and variance of 0.177.”

Q14. Table 1: Definitely a US bias. What are the implications? For example, there are cultural factors such as food and eating cultures/habits that may differ from one culture to the next; also genetic factors that may relate to ethnicity; and socio- economic and other demographic factors such as income (e.g., low income Americans cannot afford to eat healthy; work stresses and sleeping habits may differ dramatically; work environments (healthy, toxic) may differ dramatically; time/access to proper exercise and rest habits; access to medical attention… and so forth… to also include ‘genetic’ predispositions and existing medical conditions of individuals which may be related to stroke, stroke risk either directly and/or a cumulative risk increase… somewhat discussed in lines 214-220 and 229-234). Do any studies consider these factors. Petting all this “big data” demographic and cultural information into at least correlation analyses might be useful, but not entirely necessary for the purposes here. However, the authors may wish to bring up some of these concerns for future studies.

R14.

  1. We agree that authors of some included studies tried to exlude confounding factors on demographic and cultural information. However, as sayings of George E. P. Box, Professor Emeritus, University of Wisconsin (famous statistician):

“All models are approximations. Assumptions, whether implied or clearly stated, are never exactly true. All models are wrong, but some models are useful. So the question you need to ask is not "Is the model true?" (it never is) but "Is the model good enough for this particular application?"

  1. “Bayesian reanalysis of critical care trials could promote transparency and increase the rigor and reproducibility of the results of these analyses.”

Source of paper: www.atsjournals.org/doi/pdf/10.1164/rccm.202006-2381CP

Therefore, we used Bayesian statistics to confirm the association between migraine and stroke (reproducibility) and the findings in table 4 (for example “The probability that migraine increases stroke risk was 97.8%”) are new ones compare to previous meta analyses.

Q15. Lines 177-185: Good information here.

R15. Thank you for the comment.

Q16. Table 4: “Probably (%) that risk ratio…???” [risk ratio what? Need to explain more clearly what this column(s) means]. Also, forgot to mention above that terms such as “equivocal prior, skeptical prior and optimistic prior” may need rewording and brief simple definitions should be provided for each to increase readership and understanding among those less familiar with terms and expressions.

R16. We reworded “Under the three prior distributions, the results of table 4 showed risk ratio, 95% CrI, and probability of risk ratio (the probability that migraine increases stroke risk by at least 10%, 20% and 30%, respectively).”

Q17. Lines 221-228: Good points.

R17.Thanks for your comment.

Q18. Lines 229-242: Good points on some of the limitations, gaps, biases, and other considerations. This is useful to know for future research design in order to reduce some of these concerns.

R17. We appreciate your comment.

Q19. Section 5 (Conclusion) is a bit brief. More could be added, including “way forward” or recommendations for future research.

R19. We revised: “In conclusion, this current analysis shows that the risk of total stroke increased 31% among migraineurs. The probability that migraine increases stroke risk was 97.8%. The regular health examinations with focus on cerebrovascular symptoms should be a part of personal preventive health care strategy and/or and part of social welfare system. The risk of ischemic stroke or hemorrhagic stroke in migraineurs and association be-tween stroke and types of migraine (with or without aura), frequency of migraine and migraine age of onset should be further investigated in the future studies.”

Reviewer 2 Report

Overall, the meta-analysis confirms what we do and do not know about the relationship between migraines and stroke. The use of three different model types to assess the relationship is a plus.  The research makes it obvious that meta-analysis is useful and needed in research of this type.

The paper is well-written  and scientifically sound as to methods and analysis.  Criteria for inclusion/exclusion of studies are clear and appropriate.  The conclusions are consistent with the data analysis and illustrate the variability of results using different methods and a limited number of studies. The mean follow-up years is 9.86 indicating that the length of time for follow-up meets criteria for an overtime assessment.

My comments largely are related to wanting more information.   I found a need to know how “total” stroke is measured.  While it is obvious that the term refers to both types of stokes, is the measure one of the total number of strokes both ischemic and hemorrhagic?  The authors report there were almost 2.8 million participants.  Of these, about 1.4 million are from the study of children 2 – 17 years of age.  Would it make sense to exclude Gelfand’s study and rerun their analysis?  “Any age” is found in two studies- Hall and Lee.  Is it possible to be more specific?  

 I am interested in the variable “migraine” as used in the paper.  How is it operationally used?   I assume from Table 1 that “any” refers to the presence or absence of a migraine during the study periods.  Is there data on the frequency of migraines during the study periods?  Is this an issue that should be addressed in this paper for in future studies? I see that the term “stroke incidence” appears in Line 231. What does this mean?

Author Response

Q1. Overall, the meta-analysis confirms what we do and do not know about the relationship between migraines and stroke. The use of three different model types to assess the relationship is a plus. The research makes it obvious that meta-analysis is useful and needed in research of this type. The paper is well-written and scientifically sound as to methods and analysis. Criteria for inclusion/exclusion of studies are clear and appropriate. The conclusions are consistent with the data analysis and illustrate the variability of results using different methods and a limited number of studies. The mean follow-up years is 9.86 indicating that the length of time for follow-up meets criteria for an overtime assessment.

My comments largely are related to wanting more information. I found a need to know how “total” stroke is measured. While it is obvious that the term refers to both types of stokes, is the measure one of the total number of strokes both ischemic and hemorrhagic? The authors report there were almost 2.8 million participants. Of these, about 1.4 million are from the study of children 2 – 17 years of age. Would it make sense to exclude Gelfand’s study and rerun their analysis? “Any age” is found in two studies- Hall and Lee. Is it possible to be more specific?

R1. We sincerely thank the reviewer for your valuable comments.

  1. We agree with the reviewer that we need to define how to clarify “total” stroke but the authors in some papers didn’t mention specific type of stroke. So we reported “total” stroke as any type of stroke.
  2. We include Gelfand’s study because we consider association between migraine and stroke in general population, not only adults.

In classical meta-analysis, when we excluded Gelfand’s study and rerun, the conclusion unchanged (minor change in results):

RR = 1.19 [1.13; 1.24] (fixed effect/total stroke), RR = 1.31 [1.05; 1.62] (random effects/total stroke);

RR = 1.11 [1.05; 1.17] (fixed effect/ Ischemic); RR = 1.08 [0.91; 1.27] (random effects/ Ischemic);

RR = 1.19 [1.08; 1.31] (fixed effect/hemorrhagic); RR = 1.13 [0.79; 1.60] (random effect/hemorrhagic)

  1. “Any age” in 2 studies: Hall and Lee, the authors observed association between migraine and all stroke across all age and sex groups, especially 15 – 60 + years old (Hall’s study), 20 – 85+ years old. We revised “Any age” into “≥ 15” and “≥ 20” in table 1.

Q2. I am interested in the variable “migraine” as used in the paper. How is it operationally used? I assume from Table 1 that “any” refers to the presence or absence of a migraine during the study periods. Is there data on the frequency of migraines during the study periods? Is this an issue that should be addressed in this paper for in future studies? I see that the term “stroke incidence” appears in Line 231. What does this mean?

R2. We appreciate your comments.

  1. Migraine is operationally used by different methods in included papers, such as physician diagnosis, patient records, self-reported/interview.
  2. “Any” refers the presence of a migraine but we don’t have information on what type of migraine (with or without aura) in all included papers.
  3. Most authors didn’t report data on the frequency of migraines. We updated the frequency of migraines is an issue that should be addressed in this paper for future studies (last paragraph of discussion).
  4. We used “stroke incidence” instead of the risk of stroke. We deleted “incidence”

Reviewer 3 Report

The manuscript deals with an interesting topic. However, there are aspects that need to be improved. First, the introduction does not provide sufficient theoretical background to contextualise the study. The methodological section is clear, but the purpose of the study is not clearly specified. It would be interesting if the authors included research questions to manage the information found in the meta-analysis. The discussion is limited. In a manuscript of this nature it is important to organise the findings into different topics or research questions. I encourage authors to structure this section and the conclusions according to the research questions they add. It would also be interesting to address previous review studies and meta-analyses to highlight what this manuscript adds new.

Author Response

Q1. The manuscript deals with an interesting topic. However, there are aspects that need to be improved. First, the introduction does not provide sufficient theoretical background to contextualise the study.

Q2. The methodological section is clear, but the purpose of the study is not clearly specified. It would be interesting if the authors included research questions to manage the information found in the meta-analysis.

Q3. The discussion is limited. In a manuscript of this nature it is important to organise the findings into different topics or research questions. I encourage authors to structure this section and the conclusions according to the research questions they add. It would also be interesting to address previous review studies and meta-analyses to highlight what this manuscript adds new.

Response:

We thank the reviewer for your important comments.

R1. Introduction: I revised the second paragraph into “In addition, the association between migraine and stroke has also been received attention by many researchers because preventive strategies to reduce the number of stroke patients based on migraine status contribute to not only economic burden reduction of stroke for public health system but also sustainable development at the individual, community and national level.”

And “Hence, a better understanding of association between stroke and migraine in general population is needed to take steps to reduce brain health problems in the Sustainable Development Goals.”

R2. Introduction: I added “In frequentist approach, p-value is the probability of the observed outcome given that the null hypothesis is true (Ho: there is no relationship between migraine and stroke), denoted by P(data observed | alternative hypothesis), or confidence interval is inter-preted as 95% of random samples of study subjects will produce confidence intervals that contain the true proportion of migraine is associated with the risk of stroke. In contrast, Bayesian statistics aims to answer the question: Given the data we observed, what is the probability for some event of interest happens (i.e. The probability that migraine increases stroke risk at least 10%)?, which is denoted by P(event of interest | data observed).”

R3. We arranged and divided discussion into 4 parts:

Paragraph 1, 2: comparison with previous meta analyses + mechanism how migraine affects stroke

Paragraph 3: Updated findings in Bayesian meta analysis: we added “However, results in table 5 also showed that the link between migraine and any types of stroke no longer exists if 10% bias occurred in each study in the sensitivity analysis. Furthermore, the probability of migraine associated to stroke decreases significantly when bias increases from 10% to 30% in any type of stroke. This bias might be caused by inaccurate methods to ascertain migraine patients and stroke cases such as interviews or one self-reported question in included studies.”

Paragraph 4, 5: Strength and limitation, we added: “… some included cohort studies were not adequate of loss of follow up or no description of those lost which might affect the outcome results.”

Paragraph 6: Some suggestions for future studies.

Round 2

Reviewer 3 Report

The authors have made the suggested amendments. This paper has undoubtedly improved since the last revision.